# SARS-CoV-2 Transplacental Transmission: A Rare Occurrence? An Overview of the Protective Role of the Placenta

**DOI:** 10.3390/ijms24054550

**Published:** 2023-02-25

**Authors:** Yin Ping Wong, Geok Chin Tan, T. Yee Khong

**Affiliations:** 1Department of Pathology, Faculty of Medicine, Universiti Kebangsaan Malaysia, Kuala Lumpur 56000, Malaysia; 2Department of Pathology, SA Pathology, Women’s and Children’s Hospital, North Adelaide, SA 5006, Australia; 3Department of Pathology, Faculty of Health and Medical Sciences, University of Adelaide, Adelaide, SA 5000, Australia

**Keywords:** COVID-19, maternal–fetal interface, pregnancy, SARS-CoV-2, transplacental, vertical transmission

## Abstract

The outbreak of the coronavirus disease 2019 (COVID-19) pandemic, caused by novel severe acute respiratory syndrome coronavirus 2 (SARS-CoV-2), has resulted in a global public health crisis, causing substantial concern especially to the pregnant population. Pregnant women infected with SARS-CoV-2 are at greater risk of devastating pregnancy complications such as premature delivery and stillbirth. Irrespective of the emerging reported cases of neonatal COVID-19, reassuringly, confirmatory evidence of vertical transmission is still lacking. The protective role of the placenta in limiting in utero spread of virus to the developing fetus is intriguing. The short- and long-term impact of maternal COVID-19 infection in the newborn remains an unresolved question. In this review, we explore the recent evidence of SARS-CoV-2 vertical transmission, cell-entry pathways, placental responses towards SARS-CoV-2 infection, and its potential effects on the offspring. We further discuss how the placenta serves as a defensive front against SARS-CoV-2 by exerting various cellular and molecular defense pathways. A better understanding of the placental barrier, immune defense, and modulation strategies involved in restricting transplacental transmission may provide valuable insights for future development of antiviral and immunomodulatory therapies to improve pregnancy outcomes.

## 1. Introduction

Coronavirus disease 2019 (COVID-19), a viral respiratory disease caused by severe acute respiratory syndrome coronavirus 2 (SARS-CoV-2), has been declared a global pandemic by the World Health Organization (WHO) on 11 March 2020 [1]. As of 19 January 2023, there have been 663,248,631 confirmed cases of COVID-19, including 6,709,387 deaths worldwide [2].

Accumulating evidence supports that the SARS-CoV-2 virus is not merely a respiratory virus per se but can potentially affect other organ systems including the placenta. The notion of pregnancy as an altered state of immune suppression is well documented. Questions have been raised in regard to the safety and impact of SARS-CoV-2 infection on both the mothers and their unborn babies. Pregnant mothers infected with SARS-CoV-2 may be asymptomatic or symptomatic. Those who are symptomatic appear to be at a higher risk for developing severe sequelae of COVID-19 such as an increasing need for mechanical ventilation, ventilatory support, and death, compared with reproductive age-matched nonpregnant females [3]. Emerging evidence has shown that the severity of COVID-19 disease in pregnancy is associated with maternal comorbidities including gestational hypertension, diabetes mellitus, obesity, and advanced maternal age [3,4].

Like certain viral infections in pregnancies, pregnant women infected with SARS-CoV-2 are at greater risk of pregnancy complications such as premature birth and possibly cesarean delivery, which is likely related to severe maternal illness [3]. Irrespective of the emerging reported cases of neonatal COVID-19, reassuringly, vertical transmission is rare [5,6]. The majority of the neonates born to COVID-19-positive mothers were spared from getting infected, despite the SARS-CoV-2 spike protein being detected in the placental villi. Congenital viral syndromes following maternal COVID-19 infection have not yet been reported thus far [7]. A successful pregnancy is dependent on a healthy and functioning placenta. The protective role of the placenta in limiting in utero spread of the virus to the developing fetus is intriguing.

In this review, we explore recent evidence of transplacental SARS-CoV-2 viral transmission, transmission mechanics, and placental responses toward SARS-CoV-2 infection. A mechanistic description of the placental development, maternal–fetal placental interface, and innate mechanisms employed by the placenta which serves as a defensive front against SARS-CoV-2 by exerting various cellular and molecular defense pathways are also discussed.

## 2. SARS-CoV-2 and Pregnancy

### 2.1. Evidence of Vertical Transmission of SARS-CoV-2

Vertical transmission, also known as maternal-to-child transmission of SARS-CoV-2 can theoretically occur in different moments, either in utero (transplacental/congenital), intrapartum, or in the early postnatal period [8].

In utero transmission can occur transplacentally through the hematogenous route, or more rarely the ascending route from a colonized maternal genital tract. Maternal systemic viral infection may result in viremia, potentially cross the maternal–placental interface to gain access to the fetal vessels and cause infection. Unequivocal diagnosis of SARS-CoV-2 transplacental/congenital infection requires the detection of viral RNAs in the placenta tissues, fetal tissues, umbilical cord blood, and amniotic fluid [8,9] by nucleic acid amplification tests. Where a reverse-transcriptase polymerase chain reaction (RT-PCR) test to quantify viral RNA is not readily available, immunohistochemistry or an in situ hybridization technique to detect viral nucleocapsid (N) and spike (S) proteins or electron microscopy in the placental tissue may be used. In utero transmission of respiratory viruses, including SARS-CoV-2, in general are considered to be infrequent (estimated 7.7–21%) [10], except for only a few reported in utero transmission cases by the influenza virus [5,6,11]. A few studies have successfully demonstrated SARS-CoV-2 placental infection by immunohistochemistry, in situ hybridization, RT-PCR, and transmission electron microscopy techniques [12,13,14,15]. SARS-CoV-2 viral proteins, RNA, and particles were detected mostly in the syncytiotrophoblasts. Infection of other placental compartments including the decidua, cytotrophoblasts, and maternal and fetal vascular endothelial cells were also previously reported [10,12,15]. Accurate estimation of SARS-CoV-2 placental infections, however, is challenging due to the lack of standardized diagnostic criteria and consistent data collection, such as duration of maternal COVID-19 infection and viral loads [14]. Viremia due to SARS-CoV-2, although rare, appears to be associated with disease severity. Intriguingly, high SARS-CoV-2 viral load was detected in the maternal blood and the placenta from an asymptomatic mother [16]. Larger scale studies are needed to draw a definite conclusion on this issue.

Intrapartum transmission occurs during delivery and childbirth. Shedding of SARS-CoV-2 viral RNA in the feces and vaginal secretions of infected individuals, although rare, does occur. Fecal and blood contamination of the vaginal canal during the birth process could potentially expose the neonates’ oro/nasopharynx to the pathogen [17]. Nonetheless, several reports also highlighted that there was insufficient evidence to support that cesarean section was better than vaginal delivery in preventing intrapartum transmission [18,19]. Likewise, SARS-CoV-2 can be transmitted during the postpartum period through breastfeeding, direct contact to infected formites, and exposure to respiratory or other infectious maternal secretions. Data are conflicting on whether SARS-CoV-2 is present in the breast milk of infected mothers [20]; this requires further investigations.

A recent published systematic review by Musa et al., (2021) on 69 systematic reviews reported on 54,413 pregnancies infected with SARS-CoV-2 resulting in more than 30,840 neonates delivered by the infected mothers [21]. Of these, over 800 neonates were tested positive for throat swab SARS-CoV-2 RT-PCR, indicating the plausibility of SARS-CoV-2 vertical transmission from COVID-19-infected mothers. Moreover, the elevated SARS-CoV-2 IgM levels in the neonates could further support the likelihood of in utero transmission [22,23], but this is also contentious [24]. Raschetti et al., (2020) revealed in their systematic review and meta-analysis that as high as 70% of mother-to-child transmission of SARS-CoV-2 during pregnancy were likely via environmental exposure (postpartum transmission). Of the 9% confirmed vertically transmitted cases, reassuringly, only 5.7% and 3.3% were transmitted via transplacental route and intrapartum, respectively [25]. Similarly, Allotey et al., (2022) in their recent systematic review revealed that less than 2% of the babies born to mothers with SARS-CoV-2 infection tested positive for this virus via RT-PCR; the rates were even lower (1%) when restricted to babies with in utero or intrapartum exposure to the virus. The authors also reported the severity of maternal COVID-19 infection appeared to be correlated with SARS-CoV-2 positivity in the offspring [26].

Other than humans, a myriad of animal species, including dogs, cats, otters, gorillas, deer, hamsters, mink, and ferrets have been demonstrated to be susceptible to SARS-CoV-2 via natural and/or experimental infections [27]. Except for a recent report on SARS-CoV-2 experimental infection in five adult pregnant white deer with 12 fetuses, SARS-CoV-2 in pregnant animal models are generally lacking. In this report, one of the two principal SARS-CoV-2-inoculated pregnant white deer was euthanized at an acute stage of infection (day 4), and two of her three fetuses were positive for SARS-CoV-2 RNA in at least one tissue organ. Interestingly, none of the fetuses (*n* = 9) collected on day 18 post COVID-19 infection had detectable levels of SARS-CoV-2 RNA, although half of the fetuses were non-viable [28].

### 2.2. SARS-CoV-2 Cell-Entry Pathways

Viruses are obligate intracellular parasites that depend solely on hosts for survival and generate new infectious particles. Of the myriad viruses that infect human beings, only a handful can cross the placental barrier, where the resulting infections can cause fetal growth restriction, preterm delivery, or birth defects. These viruses include rubella virus, herpes simplex virus, human cytomegalovirus, Zika virus, and possibly the SARS-CoV-2 virus. Transmission of virus from the mother to her fetus in utero can occur either via maternal circulation or by ascending from the lower genital tract [29].

Maternal viremia is a prerequisite for maternal–fetal transmission by transplacental route. From maternal circulation, these pathogens can be transmitted to placental trophoblasts from infected maternal blood macrophages/vascular endothelium or through paracellular routes to fetal capillaries [30,31]. The underlying mechanisms of how these viruses can circumvent the placental defense barrier and get to the fetus has been a puzzle. Generally, viruses need to stay within the target cell in order to survive. The exact mechanism of how SARS-CoV-2 transplacental transmission occurs however is still poorly understood.

Generally, there are two main mechanisms employed by the viruses in order to gain entry into the host cell: (i) through direct fusion with the cell plasma membrane via attachment to the host cell-surface receptors or (ii) through an internalization process into endosomes and further release into the cytoplasm [32]. The tropism of viruses for the decidua and placenta depends largely on the expression of specific viral entry receptors in these tissues as well as on the maternal immune response. Consequently, infection can result in disease outcomes ranging from no effect to pregnancy loss by miscarriage or to fetal infection with subsequent congenital viral syndromes [33].

The presence of specific receptors on the plasma membrane of placental trophoblasts is a prerequisite for the entry of many viruses. It is worth noting that SARS-CoV-2 virus infects the uterine vasculature, and spreads to extravillous trophoblasts. Spike glycoprotein of SARS-CoV-2 virus interacts with the host cell surface receptor—angiotensin converting enzyme 2 (hACE2) through the receptor–binding domain. This is followed by conformational changes (priming and activation) of the spike protein by host cell proteases (such as furin, cathepsins, and transmembrane serine protease (TMPRSS-2 and -4)), to allow fusion with the host cell membrane and subsequent entry into the cytoplasm [34]. Although ACE2 receptor and TMPRSS2 are identified in placental trophoblasts, there are conflicting data with regards to the extent of their co-expression in the same cell types and whether there is differential expression by gestational age [35]. Huang et al., (2022) in their single cell RNA sequence (scRNA-Seq) analysis revealed that ACE2 receptor was similarly expressed between the lungs and the placenta, whereas TMPRSS2 was almost absent in the placenta across different stages of pregnancy. ACE2 and TMPRSS2 co-expression was seen in very few placental trophoblastic cells [36]. This was further supported by the analysis of the Human Protein Atlas, Genome-based Tissue Expression, and FANTOM5 CAGE datasets which concluded that first trimester placental tissues expressed abundance of ACE2 protein despite low mRNA levels [37].

A recent study investigated the gene expression alteration in term placentas from COVID-19 pregnant women compared to uninfected women [38]. Through robust transcriptomic analysis (microarray and scRNA-Seq) and in silico predictions of viral–host protein–protein interactions, studies revealed that almost all villous trophoblast cells, placental and trophectoderm cells express high levels of the potential non-canonical cell-entry mediators such as human dipeptidylpeptidase-4 (DPP4), cathepsin-L (CTSL) and CD147 [39,40] throughout pregnancy, and may serve as candidate binding targets of the SARS-CoV-2 spike protein. First trimester ACE2+ placental trophoblast cells and second trimester ACE2+ extravillous trophoblasts were reported to co-express CD147 and CTSL [41]. A sc-RNA expression map identified a list of 28 SARS-CoV-2 and coronavirus-associated receptors and factors (SCARFs) that may facilitate or restrict SARS-CoV-2 entry into host cells. A small population of cytotrophoblasts were found to co-express ACE2 with TMPRSS2, CD147, and/or DPP4 but exhibit very low level of interferon-induced transmembrane protein (IFITM1-3) and lymphocyte antigen 6E (LY6E) restriction factors, which could predispose a subset of the trophoblast cells to SARS-CoV-2 infection. This is consistent with the current opinion that vertical transmission from infected mother to fetus is plausible but probably rare [42]. In addition, neurolipin 1 (NRP1) is also a recently discovered alternative receptor for SARS-CoV-2 viral entry in the placenta [43]. Changes in *DAAM1* and *PAICS* gene expressions in the human placenta that encode for proteins predicted to potentially interact with SARS-CoV-2 viral proteins during pregnancy were also identified [40].

Leucine-rich repeat-containing protein 15 (LRRC15) is a novel toll-like receptor-related cell surface receptor recently discovered in the placenta, skin, lymphatics, and tissue fibroblasts. It is believed it could control viral load with antiviral and antifibrotic abilities. Unlike the ACE2 receptor, it does not act as an entry receptor for SARS-CoV-2. Mechanistically, it binds and sequesters SARS-CoV-2 virus away from ACE+ cells, and helps to suppress infection [44]. Notably, a previous study identified *LRRC15* as one of the 10 genes that was upregulated in early onset severe preeclampsia compared to the control group [45].

Via infecting the maternal immune cells, SARS-CoV-2 virions could infiltrate the placenta and transmit the virus to the fetal cells (cell-to-cell transmission) [34].

Transcytosis is another proposed mechanism to gain access to trophoblasts, as has been shown for human immunodeficiency virus (HIV) [46]. It involves movement of molecules between two cellular compartments or environments. Egloff et al., (2020) proposed that primary trophoblasts were able to transcytose opsonized or free infectious SARS-CoV-2 viral particles in an endosome-dependent manner [47]. It was also proposed that alterations in the maternal–fetal barrier due to any insults such as inflammation and ischemia could allow transmigration of SARS-CoV-2 virions throughout the placenta into the fetal environment [48]. Additional clinical and experimental studies, however, are needed to confirm these mechanisms.

### 2.3. The Effects of SARS-CoV-2 Infection on the Placenta

Following successful infection, viruses may hsve a negative impact on pregnancy outcomes through various proposed pathogenic processes, which include modulation of regulated cell death (by apoptosis, necroptosis, pyroptosis, and novel ferroptosis) and induce excessive inflammatory and repair responses.

Ferroptosis, a unique form of non-apoptotic, regulated lipotoxic cell death, has been recently implicated as a culprit for multiorgan damage in COVID-19-infected individuals. It is characterized by disproportionate iron-dependent hydroxy-peroxidation of polyunsaturated fatty acid (PUFA)-containing phospholipids in the cell membranes and a depletion of lipid peroxidation repair capacity [49]. Invading SARS-CoV-2 virus may cause cytotoxic effects to the host cells. Influx of iron following SARS-CoV-2 resulted in iron overload in the infected host cells. Iron oxidizes lipids in the Fenton reaction, a hallmark of ferroptosis which generates massive lipid reactive oxygen species (ROS) causing cell membrane damage.

Apoptosis can be induced via the extrinsic death receptor pathway and intrinsic mitochondrial pathways. Apoptotic cell death plays a critical role in host defense by effectively limiting viral expansion [50]. SARS-CoV-2-encoded accessory proteins, including ORF3a and ORF7b, can trigger apoptosis in cells by caspase-8 activation independent of BCL-2 expression (via the extrinsic apoptotic pathway) in the lung epithelial cells [51]. This process of apoptosis has been observed in other cell types including the placenta. Parcial et al., (2022) recently revealed that apoptotic-related changes were apparent in SARS-CoV-2-infected syncytiotrophoblasts and cytotrophoblasts ultrastructurally [52]. As a corollary, placental integrity and function can be compromised, instigating obstetric complications like preterm birth, pre-eclampsia, and fetal growth restriction [49].

Pathological alteration in the placenta following maternal COVID-19 infection may be attributed to the direct and indirect impact of the SARS-CoV-2 virus on the placenta. No one histopathological signature was identified in the placentas of SARS-CoV-2-infected pregnancies [53]. Apoptosis of the infected syncytiotrophoblast is recently proposed to be associated with localized SARS-CoV-2 placentitis, that is manifested histologically by a triad of trophoblast cell death, histiocyte-predominant intervillous inflammatory infiltrates, and variable perivillous fibrin deposition [54]. Fibrin deposition is a repair mechanism [55] and, when excessive, can declare as massive perivillous fibrin deposition, whereas an excessive inflammatory response can result in massive chronic intervillositis (chronic histiocytic intervillositis) [13,56]. Accumulating data suggest that SARS-CoV-2 placentitis was associated with an elevated risk of vertical transmission and adverse pregnancy outcomes [57,58,59].

COVID-19 infection may also result in maternal hypoxia leading to reduction in placental blood flow, which could be manifested indirectly in placentas as maternal vascular malperfusion (MVM). MVM features, including accelerated villous maturation, distal villous hypoplasia, villous infarction, and decidual arteriopathy were reported in SARS-CoV-2-infected placentas [13,56,60,61]. Other histopathological patterns that have been previously described in the setting of SARS-CoV-2 maternal infection include fetal vascular malperfusion and inflammatory reaction patterns such as chronic villitis, chronic deciduitis, and acute and chronic chorioamnionitis [13,56,60,61]. Notably, up to 18% of these pregnancies did not show any placental abnormalities [61]. Placental histomorphological alteration in relation to different variants of SARS-CoV-2, immunization status, timing of infection to delivery, and the rate of vertical transmission may be an area of research interest awaiting to be explored further.

## 3. Placental Structure and Defense: An Overview

### 3.1. Placental Embryology and Development

The placenta is the first and the largest chimeric organ that develops from the blastocyst following conception, made of both the fetal and maternal tissues. Its main functions are to nurture and provide nutrients to support the development of the conceptus, besides serving as physical and functional barriers against intruding pathogens and protection from maternal immune rejection to the semi-allogenic fetus [62]. Understanding the unique architecture and function of the placenta will provide a framework to address the key physical and immunomodulatory defense pathways employed by the placenta against SARS-CoV-2 transplacental infection.

The human placenta originates from the trophectoderm, the outermost layer of the blastocyst [62]. Following implantation of the embryo into the maternal endometrium (decidua), the trophectoderm differentiates into the first trophoblast lineages: early mononuclear cytotrophoblasts and the highly invasive primitive syncytium at day 8 post-conception. The latter further expands and invades into the maternal decidua, forming an expanding syncytiotrophoblastic mass. At day 9 post-conception, a system of confluent vacuoles (lacunae) appears within the syncytiotrophoblastic mass. Uteroplacental circulation is established with the formation of blood-filled lacunae following the breaching of syncytiotrophoblasts into maternal uterine capillaries at around day 12 [63].

Morphogenesis of placental chorionic villi commences at around day 10 post-conception. Rows of actively proliferating cell columns consisting of mononuclear cytotrophoblasts break through the expanding syncytiotrophoblastic mass forming primary chorionic villi [64,65]. Primary villi are transformed into secondary and subsequently tertiary villi upon migration of extraembryonic mesodermal cells forming the villous core which are then vascularized. These villi are covered by a non-proliferative and multinucleated syncytiotrophoblasts that are generated by cellular fusion of the inner progenitor villous cytotrophoblast layer.

As the pregnancy advances, cytotrophoblast cells become more sparse within the placental villi. The syncytiotrophoblasts form the only continuous epithelial layer of the chorionic villi, separating the maternal intervillous space and the fetal capillary endothelium which constitute the key interface between the mother and her fetus [66]. Notably, the human placenta is a hemochorial tissue in which the villous syncytiotrophoblast layer is in direct contact with the maternal blood; the syncytiotrophoblasts, therefore, serve as the foremost barrier against hematogenous spread of infectious pathogens [67].

Besides formation of floating villi, at day 15 post-conception, columns of proliferating cytotrophoblasts continue to expand and differentiate into extravillous trophoblast (EVT). These cells migrate from the villous tips, invade into the base of the implantation site, and anchor the placenta to the maternal decidua [68]. In addition, they invade into the wall of uterine spiral arteries resulting in vascular remodeling and subsequent transformation of these vessels into wide, low resistance conduits [69].

The placental (fetal) vasculature continues to undergo extensive expansion within the villous mesenchyme as a result of branching vasculogenesis in the late first and second trimester. As it expands, the fetal capillaries of the terminal villous core bulge against thin attenuated syncytiotrophoblasts forming vasculosyncytial membranes towards the end of pregnancy, to decrease maternal–fetal diffusion distance and thereby maximizing oxygen delivery, nutrient transport, and waste exchange with the fetus [70,71]. Failure of these processes at any stage of development could lead to placental insufficiency, compromising fetal growth and development [72,73,74,75].

The core of the chorionic villi contains several different cell types, such as Hofbauer cells (fetal macrophages), fetal endothelial cells, fibroblasts, and mesenchymal stem cells. The Hofbauer cells are found as early as day 18 post-conception and remain until term [76,77]. These innate immune cells together with fetal endothelial cells act as additional barriers to infection, which must be traversed prior to reaching fetal circulation.

### 3.2. Trophoblast Host Defense

#### 3.2.1. Immunological Barrier

The maternal–fetal interface is the direct point of contact between the mother and her fetus and governs the fetal development, regulates the maternal immune system, and safeguards the fetus from pathogens [29]. It is composed of both the maternal and fetal tissues and consists of multiple cell types. The maternal side consists of decidual stromal cells, and a wide range of decidual immune cells such as natural killer (NK) cells, macrophages, T and B lymphocytes, and dendritic cells. Decidual NK cells (70%) and macrophages (20%) are the major immune cell population at the maternal–fetal interface. As well as being engaged in immunosurveillance, decidual NK cells are mainly responsible for immune tolerance to the fetus, uterine spiral artery remodeling, and EVT cell invasion [78]. The fetal side is made of placental chorionic villus, formed by a mesenchymal core containing fetal blood vessels and Hofbauer cells, surrounded by inner villous cytotrophoblasts and outer multinucleated syncytiotrophoblasts, an epithelial covering that is in direct contact with maternal circulation [29]. Hofbauer cells can be found in villous stroma as early as three weeks post conception and are present in all stages of gestation. They are presumed to play diverse roles: they not only serve as immune cells, but are also involved in promoting and regulating placental morphogenesis and angiogenesis [79].

In addition, the paucity of toll-like receptors (TLRs) on syncytiotrophoblast confers extra syncytial resistance to infection. TLRs are the key player in activating the innate immune system that recognizes pathogen-associated immunostimulants, including peptidoglycan and lipopolysaccharide. Interestingly, TLRs are temporally expressed throughout the pregnancy. The syncytiotrophoblast layer is negative for both TLR-2 and TLR-4 in the first trimester placenta, in contrast to term placenta where TLR-2 and TLR-4 are highly expressed [80]. The lack of TLR expression by syncytiotrophoblast in early pregnancy allows placental tissues to respond only to a microbe that has penetrated through this outer layer; in other words, a pathogen can only pose a threat to the fetus if the TLR-negative syncytiotrophoblast layer is breached. On the contrary, the large apical surface of syncytiotrophoblast (12–14 m^2^) at term increases the probability of recognizing hostile microorganisms in maternal blood through TLRs [81].

#### 3.2.2. Physical Barrier

The intrinsic defense of the placenta is entrenched in its unique microarchitecture. The outermost syncytiotrophoblast cells form a fused multinucleated cell layer, lacking intercellular gap junctions that can be exploited by microorganisms through intercellular unions [82].

Syncytiotrophoblast has an unusually enriched actin cytoskeletal network that contributes to its elasticity. This cytoskeletal organization helps create a shielding brush border on its apical surface that may impede pathogen adhesion, and at the same time resists physical deformations essential for pathogen invasion [66,83,84]. Additionally, caveolins play a vital role in endocytosis and transcytosis allowing entry of some viruses into host cells. They then trigger the caveolin-1 protein system, initiate inflammatory reaction by triggering the release of cytokines such as interleukin-6 (IL-6) and tumor necrosis factor-α (TNF-α) through the nuclear factor kappa-light-chain-enhancer of activated B cells (NF-κB), leading to cell damage. Due to the low or near absence of caveolin expression in syncytiotrophoblasts, it may interfere with the success of viral transmission, including SARS-CoV-2 virus [82]. The role of trophoblast basement membrane that separates cytotrophoblasts from the villous stromal core cannot be overemphasized, as it represents an additional physical barrier to hinder effective transmission of microorganisms.

#### 3.2.3. Chemical Barrier: Cytokines, Exosomes, and microRNAs

##### Cytokines

The placenta secretes antimicrobial components that target a wide range of viruses. Interferons (IFNs) are key cytokines involved in innate host defense against pathogens, particularly RNA viruses, DNA viruses, intracellular bacteria, and parasites [85]. There are three types of IFNs namely type I IFNs (including IFN-α, β, ε, τ, and δ), type II IFN (IFN-γ) and type III IFNs (IFN-λ1, λ2, λ3, and λ4). While type I IFNs function in a broad systemic manner, type III IFNs control infection locally restricted to barrier surfaces including the maternal–fetal interface, blood-brain barrier, gastrointestinal, and respiratory epithelial surfaces.

Growing evidence suggests that type III IFNs are powerful placental gestational-age dependent antiviral proteins [86,87]. Unlike in other barrier cell types, type III IFNs are constitutively released from human trophoblasts in the absence of viral infections, without the need to undergo canonical pattern-recognition receptor (PRR)-mediated innate immune signaling pathways. Bayer et al., (2016) discovered that primary human trophoblast cells isolated from full term placentas protect placental and non-placental cells from Zika virus infection through the release of type III IFN, which functions in a paracrine and autocrine manner [88]. Likewise in murine pregnancy, placentas lacking functional type III IFN signaling demonstrated a higher rate of Zika virus vertical transmission and were associated with fetal demise and/or congenital malformations [89].

Pregnant women contracted with SARS-CoV-2 were found to have high type III IFN cytokine levels, and levels significantly increased with disease severity. It is postulated that the high level of type III IFN could be one of the possible mechanisms protecting the fetus against SARS-CoV-2 transplacental infection, although this warrants further study [90].

Similar to IFNs, NF-κB is a transcription factor that regulates inflammatory response by controlling genes that (1) are involved in the recruitment, activation, and differentiation of innate immune cells and inflammatory T cells; (2) participate in inflammasome activation; and (3) encode cytokines including interleukin 1β (IL-1β), IL-6, and tumor necrosis factor α (TNF-α) and chemokines such as IL-8, C–C motif chemokine ligand 2 (CCL2), and C–X–C motif chemokine 10 (CXCL10) [91]. NF-kB is tightly regulated in pregnancy as it plays a role in the onset of labor. Suppression of NF-κB in T cells throughout pregnancy is critical to maintain a favorable cytokine environment that is essential for the success of a pregnancy. NF-κB is downregulated in the first trimester decidua that may contribute to the immunosuppressive state during pregnancy. [92]. NF-κB has been shown to be elevated in a dose-dependent manner in response to SARS-CoV-2 viral infection. It has a pivotal role in cytokine storm syndrome, which is linked to a greater severity in COVID-19-related symptoms and could potentially serve as an attractive target in COVID-19 therapeutics [93].

Inflammatory response is a double-edged sword. These responses are crucial for protecting the developing fetus from pathogen intrusion, but at the same time, inappropriate or dysregulated expression of pro-inflammatory cytokines and IFNs can seriously disrupt placental and fetal development, leading to birth defects and pregnancy complications [90].

##### Exosomes and Trophoblastic microRNAs

Placental trophoblasts produce and release high levels of exosomes. Exosomes are regarded as small cargo vehicles that serve to transfer nucleic acids, proteins, lipids, and other biomolecules to maternal, fetal, or placental cells. Exosomes are enriched in microRNAs (miRNAs).

MiRNAs are small endogenous non-coding, single-stranded RNA that post-transcriptionally regulate the expression of a variety of target genes. MiRNAs are approximately 22 nucleotides in length and possess a longer half-life and stability that is 10 times stronger than mRNAs. They govern gene expression by mRNA degradation or by translational repression, depending upon their 3′-untranslated region (UTR) complementarity [94,95]. They serve as critical regulators in various cellular biological processes including cell proliferation, differentiation, angiogenesis, immune cell development, and apoptosis [96].

By far, more than 2000 mature miRNAs have been discovered in the human genome, among which over 600 miRNAs are identified in the human placenta [97]. A high proportion of placental-derived miRNAs originate from a large miRNA cluster located on chromosome 19 (termed chromosome 19 miRNA cluster (C19MC). C19MC contains 56 highly homologous miRNA genes within a 100 kb genomic region, and is exclusively expressed in human trophoblasts, embryonic stem cells, and some cancer cells [98]. Placental-derived miRNAs are released into the maternal circulation through encapsulation into the exosomes, and transfer their contents into other cells, mediating tri-directional communication between the mother, the placenta, and her fetus.

Studies identified at least three members of the C19MC family (miR517-3p, miR516b-5p, miR512-3p) that exhibit potent antiviral properties against RNA and DNA viruses [99]. These trophoblast-derived exosomes and C19MC-associated miRNAs attenuate viral replication by robustly inducing autophagy, conferring viral resistance to nonplacental recipient cells in autocrine and paracrine manners [99].

In addition to the C19MC family, using a bioinformatic approach, Khan et al., (2020) identified three other host miRNAs (miR-17-5p, miR-20b-5p, and miR-323a-5p) that could target SARS-CoV-2. However, the authors acknowledged that the virus may mutate; hence, the host miRNAs may change according to the new binding sites. Furthermore, the possibility of host genomic variations may also alter the miRNA binding sites [100]. Taken together, these suggest that the miRNAs might vary in different individuals depending on host genome and types of SARS-CoV-2 variants.

Interestingly, RNA viruses can create their own miRNAs which may modulate host response to the virus, by facilitating viral replication, and modulating cytokines. A study of 15 SARS-CoV-2 infected pregnant women compared to six control uninfected pregnant women identified two groups of miRNAs that were upregulated in the blood and placenta, that is, (1) seven antiviral miRNAs (miR-21, miR-23b, miR-28, miR-29a, miR-29c, miR-98, miR-326) and (2) six immunomodulatory miRNAs (miR-17, miR-92, miR-146, miR-150, miR-155, miR-223) [101]. They found that the miR-29 family has the largest number of interaction sites with SARS-CoV-2 transcripts. Notably, miR-21 was also found to be upregulated in the placentas of preeclampsia patients [102]; this is similar to the histological findings of hypertensive-like changes of the placenta in SARS-CoV-2 infected pregnant women [13]. In the case of SARS-CoV-2 virus, recent studies have suggested five plausible mechanisms employed by the host miRNAs to tune down SARS-CoV-2 infection. These include (1) binding of host cell miRNA to viral genome by translational repression or mRNA degradation and modulating viral replication; (2) regulation of host innate and adaptive immune response via modulating anti-inflammatory cytokine genes (IL-10 and TGF-1β); (3) downregulation of the expression of proinflammatory cytokine genes (IL-1β, IL-6, and TNF-α); (4) posttranscriptional regulation of the expression of ACE2 receptor and TMPRSS2; (5) interfering with SARS-CoV-2 cell entry by downregulation of ACE2 receptor and TMPRSS2 gene expression; and (6) SARS-CoV-2-derived RNA transcripts acting as competitive endogenous RNAs that may attenuate host cell miRNA expression [103,104]. Comprehensive reviews on the role of host miRNAs against SARS-CoV-2 infection were previously published [103,104]. Further investigations will provide more evidence on the antiviral properties of miRNAs and may help to create a new avenue for SARS-CoV-2 therapeutic targets which could potentially serve as an antiviral medication alternative.

Figure 1 summarizes the plausible infective pathways of SARS-CoV-2 virus into fetal circulation and various placental defense mechanisms to combat the viral infection.

## 4. Impact of Maternal SARS-CoV-2 Infection on Pregnancy Outcomes

As previously mentioned, SARS-CoV-2 entry results in placental ACE2 downregulation, which may be heightened during severe infection. Downregulation of ACE2 may lead to dysregulation of the renin–angiotensin system (RAS), resulting in elevated maternal blood pressure and dysfunctional placental vascularization. This may lead to comorbidities associated with infection in pregnancy, such as preeclampsia [105]. Mendoza et al., (2020) revealed that pregnancies with severe COVID-19 can develop clinical manifestations similar to pre-eclampsia and could be distinguishable from actual pre-eclampsia by biomarker level assessment, including serum soluble fms-like tyrosine kinase and placental growth factor [106]. This is consistent with another study from Sweden that pregnant women with COVID-19 had a higher prevalence of pre-eclampsia [107], especially those with severe COVID-19 [105].

Studies have also revealed that COVID-19 in pregnancy is associated with increased risk of preterm labor and stillbirth, especially those in the active phase of the disease or when infections occurred in the first or second trimester [108,109,110]. The preterm birth rate among COVID-19-affected pregnancies was 11.8% compared with 8.7% among those without COVID-19. Pregnancies with comorbidities and a superimposed COVID-19 infection increased the risk of preterm labor [111]. For instance, pregnancies complicated with hypertension, diabetes mellitus, and/or obesity along with a COVID-19 infection had a 160% greater risk of very preterm delivery and a 100% elevated risk of preterm delivery compared to those without comorbidities or COVID-19 infection [111]. Also, severe COVID-19-affected pregnancies were reported to have a higher incidence of spontaneous or iatrogenic preterm deliveries and preterm rupture of membranes [108]. Extreme prematurity is associated with increased mortality in the early to mid-adulthood. Many survivors may suffer from lifetime health problems, including neuropsychiatric impairment such as learning disabilities and visual and hearing problems [112]. DeSisto et al., (2021) revealed that in U.S., the stillbirth rate had increased from 0.59% (pre-pandemic stillbirth rate) [113] to 0.98% (COVID-19 affected pregnancies during pre-Delta period) and 2.70% (during the Delta period) [114]. Accumulating data suggested that impaired placental function, manifested histologically as placentitis, is a plausible mechanism of stillbirth in these cases [5,58,115].

While some studies revealed that SARS-CoV-2 infection during pregnancy was not associated with fetal growth restriction regardless of the timing of infection [13,116,117,118], lower birth weight was observed in COVID-19-infected pregnancies [105], especially when infection occurred in early pregnancy [119]. Placental fetal vascular malperfusion may be attributed to suboptimal fetal growth, especially in severe COVID-19 infection [120]. Uncomplicated deliveries and favorable neonatal outcomes were largely reported by researchers detailing first trimester COVID-19 maternal infections [121,122].

Emerging early follow-up studies reported that SARS-CoV-2 infection during pregnancy was associated with adverse neurodevelopmental outcomes in the offspring [123,124,125]. When compared with the pre-pandemic cohort, infants in the pandemic cohort were more likely to suffer from communication impairment, without significant differences in other domains such as gross motor, fine motor, and problem-solving skills [123]. Infants with exposure to SARS-CoV-2 in utero were at risk of fine motor impairment compared to those without [123]. Ayed et al., (2022) reported that only 10% of infants born to mothers with SARS-CoV-2 during pregnancies showed developmental delays. Compared to infants born to mothers who had COVID-19 infection during the third trimester, the risk of developmental delays among infants was higher in those born to mothers who had SARS-CoV-2 infection during their first and second trimesters [125]. Increased serum levels of IL-6 especially during the severe course of maternal SARS-CoV-2 infection may alter offspring’s salience network and be responsible for the subsequent cognitive impairment in the newborns, such as autism spectrum disorder, schizophrenia, and cerebral palsy [126]. Nonetheless, a definitive connection between SARS-CoV-2 exposure in utero and impaired neurodevelopment in the offspring is yet to be fully established, requiring studies with longer follow-up periods.

## 5. Concluding Remarks and Future Prospects

Ethical regulations evolving around human placental research have caused significant delay in the development of effective therapies for maternal–fetal infection. Exact mechanisms of how vertical transmission occurs in novel emerging viruses such as SARS-CoV-2 and how host–pathogen interactions in the placenta niche remain as major unresolved questions. Questions like why some but not other maternal infections result in congenital defects, what are the cellular targets of virus in the placenta, and what are the precise molecular mechanisms of viral-mediated host cell damage remain unanswered. For instance, further investigations to generate more consistent data with regards to the expression level and period of ACE2 and TMPRSS2 in the placenta in relation to different gestational ages are required. Our knowledge of LRRC15, a recently discovered potential novel biomarker that could determine SARS-CoV-2 disease severity, in the placenta and how it affects the maternal–fetal barrier in pregnancy is still limited, requiring more studies. Preeclampsia-like features that are seen in SARS-CoV-2-affected pregnancies are intriguing, with many studies demonstrating the viral affinity towards ACE2 receptor, placental histomorphological changes similar to preeclampsia, and upregulation of miRNA associated with preeclampsia in SARS-CoV-2-affected pregnancies; clear evidence of this association is still lacking. The question of whether SARS-CoV-2 can cause preeclampsia remains a mystery. Additionally, the evolutionary processes involved in the emergence of new SARS-CoV-2 viral variants that potentially result in severe illnesses in pregnancy need to be further explored. Last but not least, miRNAs, either encoded by host cells or by a viral genome, play a vital role in regulating host–cell gene expression and manipulating the cellular environment in the context of host–viral interactions. A comprehensive analysis of host and viral miRNAs and their target is warranted to provide valuable insights into the complex mechanisms underlying miRNA-mediated host–SARS-CoV-2 viral interactions during the COVID-19 pathogenesis. Future studies could address whether SARS-CoV-2 viral miRNAs have modulated the host (human) genome to enable a favorable intracellular milieu. With the advancement of next generation sequencing and artificial intelligence-powered bioinformatics, we foresee a rapid expansion in our knowledge on host–placental–viral miRNA interaction in near future, where personalized therapy could be based.

## 6. Conclusions

A better understanding of the placental barrier, immune defense, and modulation strategies involved in restricting transplacental transmission of SARS-CoV-2 and other emerging pathogens may provide valuable insights for future development of antiviral and immunomodulatory therapies to improve pregnancy outcomes.

## Figures and Tables

**Figure 1 ijms-24-04550-f001:**
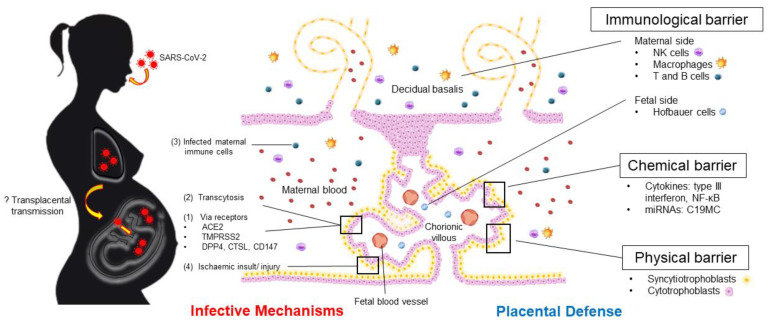
Plausible SARS-CoV-2 transplacental infective mechanisms and the various placental defense to prevent mother-to-fetal transmission. Abbreviations: ACE2: angiotensin converting enzyme 2; C19MC: chromosome 19 microRNA cluster; CTSL: cathepsin-L; DPP4: dipeptidylpeptidase-4; miRNAs: microRNAs; NK: natural killer; TMPRSS2: transmembrane serine protease.

## Data Availability

Not applicable.

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
