# Peer review of "SARS-CoV-2 Transplacental Transmission: A Rare Occurrence? An Overview of the Protective Role of the Placenta"

_ijms, 2023, doi:10.3390/ijms24054550_

Round 1

Reviewer 1 Report

The paper "SARS-CoV-2 Transplacental Transmission: A Rare Occurrence? An Overview of the Protective Role of Placenta" Wong et al. demonstrated a review of the relationship between SARS-CoV-2 and feto-maternal infection. Although this is an important field for many clinicians, several problems lessen the quality of the manuscript. The overall quality of the manuscript would be improved by a thorough revision of the overall course and appropriate proofreading.

The introduction and abstract are almost the same, and the abstract should be a summary of the whole contents.

2.2 Viral-Trophoblast Interaction: The content is superficial and does not describe the relationship between trophoblast cells and viruses as the title suggests. It should be more detailed.

2.3. The Effects of SARS-CoV-2 Infection on Placenta: The last sentence says "As a corollary, placental integrity and function can be compromised, instigating obstetric complications like preterm birth, pre-eclampsia and fetal growth restriction [39]. Clinical perinatal prognosis should be described in more detail.

Fig. 1: Hofbauer cells on the fetal side are present in the blood vessels. Is this correct?

   How many weeks of gestation does this figure represent? The place where the spiral artery flows into the intervillous space is closed by the trophoblast plug.

   The leukocytes in the figure on the defensive side of the placenta are supposed to be present in the intervillous space; however, their function in the decidua may also be important.

   Fetal blood vessels should not be located in the maternal spiral artery.

2.3. The Effects of SARS-CoV-2 Infection on Placenta: Finally, the authors wrote about the future prospects of the study, but the previous contents were superficial and did not delve deeply into the research. Challenges and Future Prospects in COVID-19 Placental Research". The contents which is described before the paragraph should be narrower and deeper and connected to this paragraph, or this paragraph should be deleted.

Since the basic contents of this manuscript are shallow, it would be easier for readers to understand if the contents are more translational, including clinical data.

Author Response

Please see the attachment:

Our Responses to Reviewer 1

The paper "SARS-CoV-2 Transplacental Transmission: A Rare Occurrence? An Overview of the Protective Role of Placenta" Wong et al. demonstrated a review of the relationship between SARS-CoV-2 and feto-maternal infection. Although this is an important field for many clinicians, several problems lessen the quality of the manuscript. The overall quality of the manuscript would be improved by a thorough revision of the overall course and appropriate proofreading.

Our response: Thank you for the comments. We’d revised the manuscript thoroughly and the manuscript was carefully proofread and reviewed by one of the co-authors, T.Y.Khong, who is a native English speaker.

The introduction and abstract are almost the same, and the abstract should be a summary of the whole contents.

Our response: Thank you for the comments. We have summarized the abstract accordingly (see page 1, line 14 - 31).

2.2 Viral-Trophoblast Interaction: The content is superficial and does not describe the relationship between trophoblast cells and viruses as the title suggests. It should be more detailed.

Our response: Thank you for the comments. We’d modified the subtitle to “SARS-CoV-2 Cell-Entry Pathways” to suit the content better (see page 4, line 146). More details on the entry pathways were also added (see page 5, lines 184 – 222).

2.3. The Effects of SARS-CoV-2 Infection on Placenta: The last sentence says "As a corollary, placental integrity and function can be compromised, instigating obstetric complications like preterm birth, pre-eclampsia and fetal growth restriction [39]. Clinical perinatal prognosis should be described in more detail.

Our response: Thank you for the comments. We’d added in the clinical data as suggested (see page 13 & 14, lines 522 - 582).

Fig. 1: Hofbauer cells on the fetal side are present in the blood vessels. Is this correct?

   How many weeks of gestation does this figure represent? The place where the spiral artery flows into the intervillous space is closed by the trophoblast plug.

   The leukocytes in the figure on the defensive side of the placenta are supposed to be present in the intervillous space; however, their function in the decidua may also be important.

   Fetal blood vessels should not be located in the maternal spiral artery.

Our responses: Thank you for the comment. We have modified Figure 1 accordingly. Also to note the fetal vessels are within the chorionic villi. A label “chorionic villous” was added to be clearer (see page 12, figure 1).

2.3. The Effects of SARS-CoV-2 Infection on Placenta: Finally, the authors wrote about the future prospects of the study, but the previous contents were superficial and did not delve deeply into the research. Challenges and Future Prospects in COVID-19 Placental Research". The contents which is described before the paragraph should be narrower and deeper and connected to this paragraph, or this paragraph should be deleted.

Our responses: Thank you for the comments. As suggested, we have improved the entire section on the future prospects which is now linked with the previous sections such as SARS-CoV-2 entry receptors, miRNA and others. In addition, we have expanded some of the previous sections by explaining the content in greater details (See page 14 & 15, line 596-619).

Since the basic contents of this manuscript are shallow, it would be easier for readers to understand if the contents are more translational, including clinical data.

Our response: Thank you for the comments. We’d added in the clinical data as suggested (see page 13 & 14, lines 522 - 582).

Thank you. We hope we’d answered all the comments satisfactorily.

Reviewer 2 Report

Dear Authors your manuscript titled "SARS-CoV-2 Transplacental Transmission: A Rare Occurrence? An Overview of the Protective Role of Placenta" is interesting and sheds light on the transmission of SARS-CoV 2 during pregnancy. The review provides useful preliminary but fundamental information on the behavior of the placenta from the immune point of view.

Regarding overview and originality: The main issue addressed is the protective role of placenta contextualized to the pandemic by SARSCoV-2. The topic is interesting; it is a review of the literature that provides information on one aspect (that of the transplacental transmission of the coronavirus) of the SARS cov-2 infection. Also allows a "review" of what are the innate immune mechanisms of the placenta and the virus-host interaction (in this case the pregnant woman).

Regarding any specific improvements: As for the methodology, the manuscript does not require further modification; it could be improved by inserting a part on animal models of virus-host infection in pregnancy if existing.

Regarding conclusion and consistency: Yes the conclusions are consistent with the purpose set by the authors and provide a future perspective on the subject and in particular on possible therapeutic and immunomodulatory approaches to be used in pregnancy.

Regarding references: Yes, the references are appropriate to the topic. Bibliographical citations are numerically consistent and up-to-date. They allow the reader to deepen further the aspects reported in the text.

Regarding further comments: The iconographic part of the manuscript allows a clearer and more direct approach to the text. In a review of this type and on this topic no further tables or graphics are required. So you have evaluated the possible transmission of SARS CoV-2 only by a  transplacental via but there are references on the transmissibility of SARS Cov2 in presence of other infections at the uterine level or that can affect the gestation period? 

Author Response

Please see the attachment:

Our Responses to Reviewer 2

Dear Authors, your manuscript titled "SARS-CoV-2 Transplacental Transmission: A Rare Occurrence? An Overview of the Protective Role of Placenta" is interesting and sheds light on the transmission of SARS-CoV 2 during pregnancy. The review provides useful preliminary but fundamental information on the behavior of the placenta from the immune point of view.

Regarding overview and originality: The main issue addressed is the protective role of placenta contextualized to the pandemic by SARSCoV-2. The topic is interesting; it is a review of the literature that provides information on one aspect (that of the transplacental transmission of the coronavirus) of the SARS cov-2 infection. Also allows a "review" of what are the innate immune mechanisms of the placenta and the virus-host interaction (in this case the pregnant woman).

Our response: Thank you for the comments.

Regarding any specific improvements: As for the methodology, the manuscript does not require further modification; it could be improved by inserting a part on animal models of virus-host infection in pregnancy if existing.

Our response: SARS-CoV-2 infection in pregnant animal models is generally lacking. After a thorough literature search, we only found one publication using pregnant white deer to study the possibility of SARS-CoV-2 vertical transmission. We had added that in (see page 4, lines 135 - 145).

Regarding conclusion and consistency: Yes the conclusions are consistent with the purpose set by the authors and provide a future perspective on the subject and in particular on possible therapeutic and immunomodulatory approaches to be used in pregnancy.
Regarding references: Yes, the references are appropriate to the topic. Bibliographical citations are numerically consistent and up-to-date. They allow the reader to deepen further the aspects reported in the text.
Our response: Thank you for the comments.

Regarding further comments: The iconographic part of the manuscript allows a clearer and more direct approach to the text. In a review of this type and on this topic no further tables or graphics are required. So you have evaluated the possible transmission of SARS CoV-2 only by a  transplacental via but there are references on the transmissibility of SARS Cov2 in presence of other infections at the uterine level or that can affect the gestation period? 

Our response: Thank you for the comment. Yes, we evaluated the possible transmission of SARS-CoV-2 only via transplacental route, without the presence of other infection. SARS-CoV-2 co-infections during pregnancy are found in our extensive literature search, regrettably, not in the context of co-infection at uterine/ placental level. We added in paragraphs describing effects of SARS-CoV-2 infection at different trimesters of pregnancies (see page 13 & 14, lines 522 - 582).

Thank you. We hope we’d answered all the comments satisfactorily.
